# Childhood Undernutrition and Its Predictors in a Rural Health and Demographic Surveillance System Site in South Africa

**DOI:** 10.3390/ijerph16173021

**Published:** 2019-08-21

**Authors:** Perpetua Modjadji, Sphiwe Madiba

**Affiliations:** School of Health Care Sciences, Department of Public Health, Sefako Makgatho Health Sciences University, Molotlegi St, Ga-Rankuwa Zone 1, Ga-Rankuwa 0208, South Africa

**Keywords:** schoolchildren, undernutrition, stunting, underweight, maternal overweight/obesity, South Africa, rural context

## Abstract

Background: Overweight and obesity are increasing at an alarming rate in South Africa, while childhood undernutrition remains persistently high. This study determined the magnitude and predictors of stunting and underweight among schoolchildren in the Dikgale and Health Demographic Surveillance System Site, a rural site in South Africa. Methods: A cross sectional study using multistage sampling was conducted among 508 schoolchildren and their mothers. Anthropometric measurements were taken from children and their mothers, while sociodemographic information was obtained from mothers using a questionnaire. The World Health Organization Anthro Plus was used to generate height-for-age and weight-for-age z-scores to indicate stunting and underweight, respectively, among the children. Maternal overweight and obesity were assessed using body mass index. Bivariate and multivariate logistic regression analyses were used to evaluate the predictors of stunting and underweight among schoolchildren. Results: Twenty-two percent (22%) of children were stunted and 27% were underweight, while 27.4% of the mothers were overweight and 42.3% were obese. The odds of being stunted were lower in younger children, whereas having a mother who was overweight/obese and had a short stature increased the odds of stunting. Access to water, having a refrigerator, and having a young mother were protective against being underweight. Having a mother who was overweight/obese increased the odds of being underweight. Conclusions: The study showed a high prevalence of stunting and underweight among children, and overweight and obesity among mothers, indicating a household double burden of malnutrition. The age of the child and maternal overweight/obesity and short stature were predictors of stunting and underweight, while having a younger mother and access to water and a refrigerator were protective against being underweight. The need for an evidence-based and feasible nutrition program for schoolchildren, especially those in rural schools, cannot be over-emphasized.

## 1. Introduction

Childhood undernutrition remains a major public health problem in Sub-Saharan Africa, including South Africa [1,2]. Undernutrition implies having a low weight for one’s age (underweight), low weight for height (wasting), being too short for one’s age (stunting), and/or a vitamin and mineral deficiency [3,4]. Prolonged undernutrition impairs cognitive and physical development and predisposes the development of cardio-metabolic disease that may be transmitted to the next generation [5,6]. Furthermore, undernutrition predisposes children to poor achievements at school, lower productivity, and a greater risk for poor health outcomes that reduce the quality of life [7,8,9,10,11].

Underweight (i.e., low weight-for-age) remains one of the most common causes of morbidity and mortality among children throughout the world and is a composite measure of stunting and thinness (i.e., wasting) [12,13]. Underweight is an indicator for assessing changes in the magnitude of malnutrition over time [12,13]. In low-to-middle income countries (LMICs), 34–62% of schoolchildren are underweight [14]. In South Africa, national nutritional surveys estimate the prevalence of underweight among children to be 9–12% [15,16,17]. Factors predicting underweight in LMICs are poverty, household food insecurity, inadequate intake of nutrients, poor childcare practices, and unhealthy living environments [18,19,20].

The United Nation International Children’s Fund (UNICEF) has indicated that stunting is the most prevalent form of child undernutrition. Globally, more than 200 million school-aged children are stunted and underweight, and if no action is taken, the number will grow to nearly a billion by 2020 [21]. Stunting is a good indicator of long-term undernutrition and serves as a marker of multiple pathological disorders associated with increased morbidity and mortality [22]. According to UNICEF reports, stunting affects 52% of school-aged children in LMICs. The majority of stunted children in the world (90%) live in Africa and Asia [23]. South Africa is one of 24 high-burden countries that account for 80% of the world’s stunted children [21]. National nutritional surveys conducted within the past two decades in South Africa found that stunting was the most common form of undernutrition among children. These surveys reported that the prevalence of stunting ranged from 11–27% [15,16,17]. However, a recent review of studies conducted among children in South Africa reported a prevalence of stunting of 16%, 26.5%, and 30.6%, in urban, rural, and commercial farming settings, respectively [24]. 

The World Health Organization (WHO) Framework on Childhood Stunting indicates that stunting results from a complex interaction of household, environmental, socioeconomic, and cultural influences [25]. Stewart et al. [25] identified layers of contextual and causal factors that led to stunted growth and development, for example; economy, food systems, water, and sanitation. They further reported that community and societal conditions underlie infant and young child feeding practices, which are a central pillar for healthy growth and development [25]. In South Africa, undernutrition is attributable to poverty, which compromises food security. One in five households are affected by compromised food security, many of which are located in rural communities [15,26,27]

School-aged children are amongst the most vulnerable groups within the population due to their susceptibility to undernutrition, which is attributable to the vicious cycle of poverty [10,28,29]. In addition, the vulnerability of schoolchildren to undernutrition is due to their low dietary intake, decreased access to food, inequitable distribution of food within households, improper food storage and preparation, dietary taboos, and pathogenic infections [10]. The primary school age period is an important nutritional period during which the body builds up nutrition stores in preparation for the rapid growth indicative of adolescence [7,30]. Growth spurts, which occur in school going children, are nutritionally demanding and require increased calories and nutrient intake [31]. Inadequate calories and nutrient intake during school age predisposes children to undernutrition [7,8,9,10,11]. 

In South Africa, undernutrition remains persistent among schoolchildren, despite efforts to alleviate it through the National School Nutrition Program, which was introduced to provide primary schoolchildren with nutritious meals. Stunting, in particular, is identified as a major global health priority [32] because of its association with economic, developmental, and health outcomes [22,33]. However, not much research has been conducted on the magnitude and predictors of stunting and underweight among schoolchildren in rural areas in South Africa. Sustainable Development Goal two aims to end all forms of hunger and malnutrition by 2030 and ensure that children have sufficient and nutritious food all year round [34]. This study serves as a baseline indicator of the magnitude and predictors of stunting and underweight among primary school children in a rural Health and Demographic Surveillance System Site in the Limpopo Province, South Africa. The study findings will further inform nutritional interventions for catch-up growth among schoolchildren.

## 2. Methods

### 2.1. Study Design

This paper is an extraction from a doctoral dissertation written by the first author, which determined the growth patterns of primary school children and the maternal factors influencing these growth patterns. The doctoral dissertation also explored the influence of the cultural beliefs and practices of mothers on the growth of children in the Dikgale Health and Demographic Surveillance System Site. The study used a convergent mixed method design with parallel phases of quantitative and qualitative enquiry. A cross-sectional quantitative survey was used to determine the growth patterns of schoolchildren using nutritional indicators for stunting, underweight, thinness, and overweight/obesity. In addition, data on the anthropometry, socio-demographics, obstetric history, knowledge of nutrition and child growth, the influence of societal cultural beliefs and practices on child nutrition and food security were collected from the mothers. In the qualitative phase of the enquiry, focus group discussions were conducted to explore the influence of socio-cultural beliefs and practices of the mothers on their children’s growth and nutrition. The study was conducted from August 2017 to December 2017. This paper reports on the prevalence and predictors of stunting and underweight among schoolchildren in the research population in a rural context.

### 2.2. Study Setting

The study was conducted in the Dikgale Health and Demographic Surveillance System Site (DHDSSS). The DHDSSS is a well-researched rural site that was founded in 1995 and forms part of the International Network for the Demographic Evaluation of Populations and their Health (INDEPTH). INDEPTH is an umbrella organization for a group of independent health research centers operating 43 Health and Demographic Surveillance Sites in 20 LMICs [35]. The DHDSSS is situated approximately 40 km northeast of Polokwane, the capital city of the Limpopo Province, in South Africa. The area comprises of communities clustered in 16 villages with a population of approximately 36,000 with poor infrastructure. Electricity and mobile phone networks are found everywhere, while the supply of piped water is more problematic [36]. A poor socio-economic status, characterized by high unemployment and poverty, has been reported in this area [37,38].

There are 19 public primary schools in the villages forming part of the DHDSSS, with an estimated total enrolment number (EN) of learners of 7772 in 2016, ranging from an enrollment number of 112 children in the smallest school to 776 in the largest school [39]. The primary schools in this area belong to quintile three (Q3), which in South Africa are declared as no-fee schools, and therefore do not charge school fees. These schools receive the majority of their funding from the government [40]. In addition, primary schools belonging to this site have a feeding program to provide learners with meals during school hours. Although the DHDSSS is a well-researched site, there is a paucity of data on the nutritional status of children in this area.

### 2.3. Study Participants

This was a child–mother paired study. The study population comprised of primary school learners and their mothers.

### 2.4. Sample Size and Sampling Procedure

The sample size was calculated using Rao software [41]. The software takes into consideration the population size, a 5% margin of error, a 95% confidence level, and a 30% non-response rate. A total of 508 child–mother pairs were taken as the sample size. A multistage sampling technique was used. First, the schools were stratified by the size of enrollment and five of the largest schools were selected. Second, in each selected school, one class per grade was randomly selected. Third, all learners in the selected class were included. The study excluded children who were younger than five years, had physical disabilities that compromised their stature, or whose biological mothers were not available to participate.

### 2.5. Data Collection

A structured interviewer-administered questionnaire, translated from English to a local language (Sepedi), was used to collect data. The questionnaire took into consideration the determinants of nutritional status [42] and covered a range of topics on socio-demographics and the household situation of mothers, in accordance with the variables used in other studies conducted in the study area [37,43]. The questionnaire was pre-tested in a pilot study and four trained research assistants were employed to collect the data. The anthropometry (weights and heights) of the children and their mothers was recorded using a well-calibrated, smart D-quip electronic scale and a height measuring board, respectively. Height was measured to the nearest 0.1 cm and weight to the nearest 0.1 kg. All measurements were taken three times, and the average recorded. A non-stretchable plastic tape was used to measure the waist and hip circumferences of the mothers, which were recorded to the nearest 0.1 cm. All measurements were done according to WHO recommendations [44,45].

For the children, anthropometric measurements were converted to height-for-age z scores (HAZ) and weight-for-age z scores (WAZ) and compared to reference data for 5–19 year olds. The children were classified as stunted if the HAZ was less than or equal to −2SD or underweight if the WAZ was less than or equal to −2SD. The Anthro-plus software was unable to generate weight-for-age (WAZ) values for 189 children because the indicator excludes children aged 11 years and above. Thus, a sample of 319 was analyzed for WAZ for children 10 years old and younger. According to the software, WAZ reference data are not available beyond 10 years of age because this indicator does not distinguish between height and body mass in the age period where many children experience pubertal growth spurts and may appear to have excess weight (by weight-for-age) when in fact they are just tall [45].

For the mothers, body mass index (BMI) was calculated as the weight in kilograms divided by the height in meters squared (BMI (kg/m^2^) = weight (kg)/height (m^2^)). Normal BMI is within 19 to 24 kg/m^2^. Underweight is defined as BMI < 18.5 kg/m^2^, overweight as BMI of 25 to 29.9 kg/m^2^, and obesity as BMI ≥ 30 kg/m^2^. The cut-off point for central obesity in females is a waist circumference ≥88 cm [46]. The waist–hip ratio (WHR) was computed as the waist circumference divided by the hip circumference. The WHR cut-off point (i.e., abdominal obesity) for females is ≥0.85 [46].

### 2.6. Statistical Analysis

The data were analyzed using STATA version 14. Descriptive statistics for the age, body weight (W), height (H), and HAZ and WAZ of the children were computed for the mean, the standard deviation (SD), the median, and the interquartile range (IQR). Comparison of the means was done using a Mann–Whitney test, while the percentages of children with variables below, on, or above the cut-off points were compared using a chi-square test. Bivariate and multivariate logistic regression analysis was used to determine the association between the nutritional status indicators of children, their stunting and underweight, and independent variables. Bivariate analyses were used to identify the association between the dependent variables and each of the independent variables. Independent variables that had a *p*-value of 0.1 were used in the multivariate logistic regression with a stepwise backward elimination procedure controlling for confounding. During multivariate logistic regression analysis, child gender, learning grade, maternal age, WHR, WC, marital status, employment, education, household income, and household size were controlled to determine the association of stunting with covariates. For underweight, child age and gender, learning grade, maternal WHR, WC, height, marital status, employment, education, household income, and household size were controlled. Adjusted odds ratios (AOR) with a 95% confidence interval (CI) were generated and used to determine the independent strength of the associations. Significance was considered at *p* < 0.05.

### 2.7. Ethics Statement

This study was conducted according to the guidelines laid down in the Declaration of Helsinki and all procedures involving human subjects were approved by Sefako Makgatho Health Sciences University Research and Ethics Committee (SMUREC) (SMUREC/H/161/2016: PG). Furthermore, this study received permission from the Department of Education (DoE) in the Limpopo Province, South Africa. The nature of the study was explained to the mothers of the children prior to their participation. Informed consent was obtained from the mothers and verbal assent was obtained from the children.

## 3. Results

### 3.1. Distribution of Schoolchildren

A representative sample of 508 primary schoolchildren aged 6–15 years was achieved in this study, with a mean age of 10 ± 2.2 years. Of the 508 child participants, 209 (41%) were boys and 299 (59%) were girls. The children who participated in this study were in grades 1 to 7, and were categorized into two phases by school grade. The foundational phase was categorized as school grades 1 to 3 (*n* = 225) and the intermediate phase as school grades 4 to 7 (*n* = 283). The children were further divided into two age groups; younger children aged 6–9 years old (*n* = 254) with a mean age of 8 years (SD = ±1) and older children aged 10–15 years old (*n* = 254) with a mean age of 11 years (SD = ±1). The anthropometric data from 508 schoolchildren were imported into Anthro-plus to generate HAZ and WAZ. From the sample of 508 children, the software used a weighted sample of 508 for HAZ (i.e., all children) and 319 for WAZ (i.e., children who were 10 years old and younger). This meant that 189 children aged above 10 years and older were excluded by the software and, thus, had no WAZ values.

### 3.2. Nutritional Indicators of Children

In Table 1, the mean for age, weight, height, HAZ, and WAZ of the children were compared by school phase, age group, and gender using the Mann–Whitney test. Significant differences in age (*p* ≤ 0.0001), weight (*p* ≤ 0.0001), and height (*p* ≤ 0.0001) were observed between the foundation and intermediate phases. No significant differences in HAZ (*p* ≤ 0.25) or WAZ (*p* ≤ 0.26) were observed. Significant differences in age (*p* ≤ 0.0001), weight (*p* ≤ 0.0001), height (*p* ≤ 0.0001), HAZ (*p* ≤ 0.0001), and WAZ (*p* = 0.03) were observed between children aged 6–9 years and 10–15 years. The results showed no significant differences in age, weight, height, HAZ, or WAZ between boys and girls.

The comparison of the prevalence of stunting and underweight by the categorical independent variables are presented in Table 2. The prevalence of stunting in schoolchildren in this study was 22%. A significant difference in the prevalence of stunting was observed between the foundation and intermediate phases (16.6% vs. 26.2%, *p* = 0.01), and between younger children and older children (13.9% vs. 29.6%, *p* ≤ 0.0001). There was no significant difference observed between boys (25%) and girls (20%, *p* = 0.14). Severe stunting (HAZ < −3 z-scores) was observed in 5% of children with stunting. A few (10; 2%) children were very tall according to the HAZ cut off measures and were excluded from the analysis. The prevalence of underweight in the schoolchildren was 27%. There was no significant difference between the foundation and intermediate phases (27% vs. 28%), younger and older children (25% vs. 35%), or between boys and girls (28% vs. 27%). Of the underweight children, 6% were severely underweight (WAZ < −3 z-scores). A small proportion (38; 11.9%) of children had a growth problem as indicated by the WAZ cut off measures and were excluded from the analysis. Just less than half (43.4%) of the children suffered a combination of underweight and stunting (results not shown in tables).

### 3.3. General Characteristics of Mothers

Table 3 presents the general characteristics of the mothers. Slightly over half of the mothers (285; 56%) who participated in this study were aged 35 years and above, while 223 (44%) were below 35 years of age. Over one-quarter (139; 27.4%) of the mothers were overweight and 215 (42.3%) were obese. Central obesity, characterized as a waist circumference ≥ 88 cm, was observed in 269 mothers (52.9%), while abdominal obesity, characterized by a WHR ≥ 0.85, was observed in 168 mothers (33.1%). Most of the mothers (321; 63.2%) were single, had obtained a high literacy (i.e., completed secondary school and/or tertiary education) (302; 59.4%), were unemployed with no monthly income (418; 82.3%) and were living on social grants (441; 86.8%) as their source of income. One hundred and seventy (170; 33.5%) mothers were the head of the household. The majority of mothers were from large households with 5–9 family members (293; 57.7%). Most of the houses were made of bricks (319; 63.2%) and 399 (78.5%) had electricity as a source of energy. Refrigerator use was reported by 426 (83.9%) mothers and cooking using electricity was common (399; 78.5%). Water was accessible in 375 (73.8%) households and the use of pit toilets was reported by 486 (95.7%) mothers.

### 3.4. Predictors of Stunting and Underweight

The results of logistic regression analyses to determine the predictors of stunting and underweight are presented in Table 4. After controlling for child gender, learning grade, maternal age, WHR, WC, marital status, employment, education, household income, and household size, stunting was significantly associated with the child’s age, maternal height, and maternal BMI. Younger children (6–9 years old) were less likely to be stunted (AOR = 0.37, 95%CI: 0.23–0.59, *p* ≤ 0.0001). Mothers who were short were two times more likely to have stunted children (AOR = 2.1, 95%CI: 1.34–3.56, *p* = 0.003) and mothers who were overweight/obese were almost two times more likely to have stunted children (AOR = 1.90, 95%CI; 1.25–2.78 *p* = 0.002). On the other hand, underweight was significantly associated with maternal age, maternal BMI, water access, and having a refrigerator, after controlling for child age and gender, learning grade, maternal WHR, WC, height, marital status, employment, education, household income, and household size. Younger mothers were less likely to have underweight children (AOR = 0.48, 95%CI: 0.28–0.83, *p* = 0.009). Mothers who were overweight/obese were almost two times more likely to have underweight children (AOR = 1.93, 95%CI: 1.18–3.15, *p* = 0.009). Children with access to water were less likely to be underweight (AOR = 0.48, 95%CI: 0.24–0.99, *p* = 0.05) when compared to children with no access to water (AOR = 0.48, CI: 0.24–0.99). Children living in a household with a refrigerator were less likely to be underweight (AOR = 2.22, CI: 1.07–4.60).

## 4. Discussion

This study determined the prevalence and predictors of stunting and underweight in primary schoolchildren in a rural context. Most of the mothers and their children lived within a depressed socio-economic status, as indicated by high rates of unemployment with no income (82.3%) and a dependency on social grants (86.8%). The majority lived in large households with 5–9 members, used pit toilets (95.5%), and had access to water (73.8%), with electricity being a common source of energy (78.5%). The South African National Health and Nutrition Examination Survey (SANHANES-1) reported unfavorable socio-economic conditions in most South African households [16]. Mothers in the current study were overweight (27.4%) or obese (42.4%), which is consistent with findings of earlier studies conducted in the DHDSSS. These studies showed a high prevalence of overweight, obesity, and micronutrient deficiencies in women of reproductive age [37,43,47]. As early as 2002, pockets of high overweight (27%) and obesity (32%) prevalence were reported among black South African women in other parts of the country [48,49].

In the current study, the prevalence of undernutrition in the form of stunting (22%) and underweight (27%) was reported among the child participants. In contrast with the reports of studies in other rural South African settings, this study reports a higher prevalence of stunting compared to the previously estimated 6–9% [27,50] and 7.4–14.9% [51,52]. However, a similar prevalence of stunting in schoolchildren to that observed in the current study has been reported in Kenya (24%) [53]. A higher prevalence of stunting among schoolchildren than that reported in the current study has been reported in Zambia (28.9%), Ethiopia (37.9%), and Indonesia (46.5%) [54,55,56]. In the current study, the children in the intermediate phase of school were significantly more stunted than the children in the foundation phase (26% vs. 16%). In addition, more boys were stunted in comparison to girls (25% vs. 20%), although the difference was not significant. Inadequate energy intake contributes to the higher level of stunting in boys, mainly because boys above 10 years of age require more food than girls of the same age [53].

The current study reported the age of the child, maternal height, and maternal BMI as predictors of stunting. A higher prevalence of stunting was observed among children aged 10–15 years (30%) than children aged 6–9 years (14%). The study reports reduced odds of younger children being stunted (AOR = 0.21, CI: 0.09–0.48). The findings of this study are consistent with those reported in various parts of Ethiopia and in Indonesia; that stunting is significantly higher in children older than 10 years when compared to younger children [55,56,57,58]. In the current study, the high prevalence of stunting observed may be due to household poverty, as indicated by the poor socio-economic status reported in the majority of households. Similar to the suggestions made by other studies [56,59], this study attributed stunting to a chronic nutritional problem, which develops over a relatively long period. This study showed that children born to mothers whose height was ≤1.55 m were more likely to be stunted (AOR = 2.2, CI: 1.34–3.69) than children born to mothers whose height was >1.55 m. This association between a mothers’ short stature and stunting in children has been reported in other studies [60,61].

In the current study, mothers who were overweight/obese were more likely to have stunted children (AOR = 1.90, CI: 1.27–2.84). Similar findings on the association between stunting and maternal BMI has been reported in India [62]. The results indicate a dual-burden of overweight/obesity in women and the prevalence of stunting in children. Similar observations have been reported [63]. The body of literature on nutrition transition and the double burden of malnutrition points to changing diets as a contributing factor to mothers who are overweight/obese and stunted children [64]. A household diet that favors energy-dense food is unlikely to be adequate for child growth and adult health. Moreover, a higher energy intake alone is probably ineffective in preventing child stunting and may increase the risk of obesity in adulthood [65].

This study further showed a prevalence of 27% for underweight in schoolchildren. There were no significant differences in the prevalence of underweight for gender, age groups, and school phases. In contrast with the reports of studies in other rural South African settings, this study reported a higher prevalence of underweight (27%) compared to the 5–7% [27] and 4% [50] reported in rural settings and 5.45% [51] and 15% [66] reported in urban areas in South Africa. A higher prevalence of underweight than that reported in the current study has been observed in countries such as Ethiopia (40.2%) and in Indonesia (75.3%) [67,68]. Underweight is an overall indicator of the population’s nutritional health and children in rural areas are more affected by underweight than those in urban areas [24,68]. The high prevalence of underweight in children living in rural areas is associated with food insecurity and household poverty [69], which is the case in the present study.

A significant association of underweight with maternal age, BMI, access to water, and having a refrigerator was observed in this study. Factors associated with underweight among children below five years of age have been well studied in various countries such as Ghana, Ethiopia, and Nepal [70,71,72]. However, there is a paucity of data on the predictors of underweight among schoolchildren in South Africa and other LMICs. This study showed that younger mothers, below 35 years of age, were less likely to have underweight children (AOR = 0.48, CI: 0.28–0.83). This is similar to a study in Rwanda that showed that mothers who were over 35 years of age were more likely to have underweight children compared to those below 35 years of age [18].

In the current study, underweight predictors were maternal age, BMI, access to water, and having a refrigerator. In addition, overweight/obese mothers were more likely to have underweight children (AOR = 1.9, CI: 1.18–3.15). The results further showed that not having a refrigerator was significantly associated with being underweight (AOR = 2.22, CI: 1.07–4.60). The current study also revealed that access to water was protective against being underweight (AOR = 0.48, CI: 0.24–0.99). In African countries such as Nigeria, the probability of a child being underweight was significantly lower for children drinking borehole or piped water. Researchers have reported that bad-quality water can predispose diarrhea, which is associated with underweight [73].

## 5. Limitations

The findings from this study should be considered in view of some limitations. Similar to other studies using a cross-sectional study design, we could only make inferences about the associations of stunting and underweight with child and maternal factors. We could not establish causality or temporality of events, which requires a longitudinal study design. Nevertheless, we believe that the present study sheds light on the prevalence and predictors of stunting and underweight among school-aged children in the DHDSSS, South Africa. The results of this study cannot be generalized to other areas in South Africa, since this site is a very small rural area in one specific province and the situation may vary considerably in urban areas.

## 6. Conclusions

In conclusion, the prevalence of stunting and underweight were high among schoolchildren while their mothers were overweight and obese, indicating a household double burden of malnutrition. Thus, the need for an evidence-based and feasible nutrition program for primary school children, especially those in rural schools, cannot be over-emphasized. The study also found that the child’s age, maternal stature and, BMI were significant predictors of stunting, while maternal age, maternal BMI, access to water, and not having a refrigerator were predictors of underweight. This study advocates a dedicated effort by the South African government to further address stunting and underweight among schoolchildren in rural areas, toward achieving goal two of the Sustainable Development Goals, while considering ongoing school nutrition services. Furthermore, public health campaigns on the consumption of balanced diets that involve nutritionists at schools and the education of mothers to enhance the acceptability of health interventions are paramount to improve the nutritional status of children.

## Figures and Tables

**Table 1 ijerph-16-03021-t001:** The means of age, anthropometry, and nutritional indicators of schoolchildren by school phase, age group, and gender.

Variables	All Median (IQR)	Foundation Phase Median (IQR)	Intermediate Phase Median (IQR)	*p*-Value
Age (years)	10 (8; 11)	8 (7; 8)	11 (10; 12)	≤0.0001
Weight (kg)	31.1 (22.7; 36)	24.3 (20.5; 26)	36 (29.9; 42.3)	≤0.0001
Height (cm)	136 (124; 145)	125 (119; 130)	144 (136; 151)	≤0.0001
HAZ	−0.1 (−0.9; 0.6)	−0.1 (−0.8; 0;8)	−0.1 (−1; 0.6)	0.25
WAZ *	−0.2 (−1.0; 0.4)	−0.3 (−0.1, 0.3)	−0.1 (−0.9; 0.6)	0.26
		**6–9 years**	**10–15 years**	
Age (years)	10 (8; 11)	8 (7; 9)	11.5 (10; 13)	≤0.0001
Weight (kg)	31.1 (22.7; 36)	24.7 (20.6; 26.8)	37.3 (30.1; 43.8)	≤0.0001
Height (cm)	136 (124; 145)	126 (119; 131)	145 (137; 152)	≤0.0001
HAZ	−0.1 (−0.9; 0.6)	0.2 (−0.8; 0.8)	−0.3 (−1.13; 0.5)	≤0.0001
WAZ *	−0.2 (−1.0; 0.4)	−0.1 (−0.8–0.4)	−0.6 (−1.3; 0.04)	0.03
		**Boys**	**Girls**	
Age (years)	10 (8; 11)	10 (8; 11)	10 (8; 12)	0.78
Weight (kg)	31.1 (22.7; 36)	30.1 (22.7; 33.3)	31.7 (22.7; 39.2)	0.29
Height (cm)	136 (124; 145)	135 (124; 144)	136 (125; 147)	0.22
HAZ	−0.1 (−0.9; 0.6)	−0.2 (−0.9; 0.6)	−0.04 (−0.9; 0.7)	0.38
WAZ *	−0.2 (−1.0; 0.4)	−0.3 (−0.9, 0.3)	−0.2 (−0.1; 0.4)	0.55

* WAZ was only calculated for children aged 10 years and younger.

**Table 2 ijerph-16-03021-t002:** Comparison of the prevalence of stunting and underweight in schoolchildren by categories.

Categories	Normal *n* (%)	Stunted *n* (%)	*p*-Value
All Children	389 (78)	109 (22)	
Foundational Phase (*n* = 223)	186 (83)	37 (17)	0.01
Intermediate Phase (*n* = 275)	203 (74)	72 (26)	
Younger Children (*n* = 245)	211 (86)	34 (14)	≤0.0001
Older Children (*n* = 253)	178 (70)	75 (30)	
Boys (*n* = 209)	155 (75)	52 (25)	0.14
Girls (*n* = 291)	234 (80)	57 (20)	
	**Normal**	**Underweight**	
All Children	204 (73)	77 (27)	
Foundational Phase (*n* = 193)	141 (73)	52 (27)	0.80
Intermediate Phase (*n* = 88)	63 (72)	25 (28)	
Younger Children (*n* = 218)	163 (75)	55 (25)	0.13
Older Children (*n* = 63)	41 (65)	22 (35)	
Boys (*n* = 119)	86 (72)	33 (28)	0.92
Girls (*n* = 162)	118 (73)	44 (27)	

**Table 3 ijerph-16-03021-t003:** General characteristics of mothers (*n* = 508).

Variables	Categories	Frequency (*n*)	Percentage (%)
Age (years)	<35	223	44
	≥35	285	56
BMI (Kg/m^2^)	Normal	142	27.9
	Underweight	12	2.4
	Overweight	139	27.4
	Obesity	215	42.3
Waist Circumference (cm)	Normal	239	47.1
	Central Obesity	269	52.9
WHR	Normal	340	66.9
	Abdominal Obesity	168	33.1
Marital Status	Ever Married	187	36.8
	Single	321	63.2
Level of Education	Low Literacy	206	40.6
	High Literacy	302	59.4
Employment	Employed	90	17.7
	Unemployed	418	82.3
Occupation	None	418	82.3
	Civil Servant	16	3.1
	Labor/domestic Work	74	14.6
Child Support Grant	Yes	441	86.8
	No	67	13.2
Type of House	Bricks	319	63.2
	RDP/Mud/shacks	189	36.8
Household Head	Self	170	33.5
	Spouse/partner	193	38.0
	Parents	117	23.0
	Grandparents	19	3.7
	Relatives	9	1.8
Household Income	≤R1000	180	35.4
	R1001–R5000	261	51.4
	R5001–R10,000	67	13.2
Number of Household Members	1 to 4	183	36.0
	5 to 9	293	57.7
	>10	32	6.3
Number of Children in the Household	1	30	7.5
	2	132	24.4
	≥3	346	68.1
Source of Energy	Electricity	399	78.5
	Other	109	21.5
Refrigerator Use	Yes	426	83.9
	No	82	16.1
Access to Water in Yard	Yes	375	73.8
	No	133	26.2
Type of Toilet	Flush toilet	22	4.3
	Pit toilet	486	95.7

**Table 4 ijerph-16-03021-t004:** Multivariate analysis of predictors for stunting and underweight in schoolchildren.

Stunting *	AOR	95%CI	*p*-Value
Child’s age	0.37	0.23–0.59	≤0.0001
Maternal height	2.12	1.34–3.56	0.003
Maternal BMI	1.90	1.25-2.78	0.002
**Underweight ****			
Maternal age	0.48	0.28–0.83	0.009
Maternal BMI	1.93	1.18–3.15	0.009
Access to water	0.48	0.24–0.99	0.05
Access to a refrigerator	2.22	1.07–4.60	0.03

* Controlled for child gender, learning grade, maternal age, WHR, WC, marital status, employment, education, household income, and household size; ** Controlled for child age and gender, learning grade, maternal WHR, WC, height, marital status, employment, education, household income, and household size.

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
