# Peer review of "Childhood Undernutrition and Its Predictors in a Rural Health and Demographic Surveillance System Site in South Africa"

_ijerph, 2019, doi:10.3390/ijerph16173021_

Round 1
Reviewer 1 Report
Title: The title suggests the study is representative of South Africa while the description in the methods indicate a specific rural site. The title needs to reflect that this study is about the predictors of undernutrition in that rural site in South Africa rather than generalizing to the whole population.
Line 33: The definition of undernutrition in this line excludes wasting (low weight for height) which is also an indicator of childhood undernutrition. This indicator was also not assessed by the authors in their study. Is there a reason for leaving this out?
Line 152: Please indicate the reference for waist circumference cut off.
Line 201: What is the definition of severe stunting and severe underweight?
Table 1: Title should be modified to indicate that the table content are anthropometric values.
The Anthro software did not assess the Z-scores for children above 10 years old according to lines 142 to 148. How did the authors come up with HAZ and WAZ values in this table for the 10-15 years age group?
Table 2: In the third column change “stunting” to “stunted”.
Table 3: Under “number of household members”, percentages do not sum up to 100 and no explanation is given. Also, what statistical test was used to compare the percentages and where are the p-values?
Table 4: Authors need to indicate what variables were adjusted for in the regression analysis. This should be indicated under the table, and also mentioned in the text instead of stating it simply as “after controlling for independent variables” in line 225.
Line 225: Indicate what independent variables were controlled for.
Line 274: “Overweight/obesity mother” does not read well. Authors may want to change to “mothers with overweight/obesity”
Line 287: It is not clear how 40.2% and 75.3 % are lower prevalence rates than the 27 % found in the current study. Do the authors mean higher rates instead?
Author Response
All the corrections and additional information are written in blue in the manuscript:
The title has been changed to reflect that the study is about predictors and the study area has been added: Line 2 to 4.
The definition of undernutrition included wasting (i.e. low weight for height) in Line 38. Also, we did not including wasting in this study because BMIZ is reported in first paper submitted to BMC Public Health on Thinness and Overweight and obesity as a nutritional indicator. Hence, this paper has excluded BMIZ for children and focused on stunting (low HAZ) and underweight (low WAZ) among children, a pattern used in many papers published in international journals.
The reference for waist circumference has been added in Line 165.
The definitions for severe stunting and severe underweight have been included in Line 222 and line 227, respectively.
Table 1 in line 261, has been labelled as advised. However we had to include age and nutritional indicators in addition to anthropometry to distinguish the variables. Not all variables in table 1 are anthropometry.
WHO Anthroplus did not generate the values for WAZ for children aged above 10 years due the reason given by the software, which is "WAZ reference data are not available beyond age 10 years because this indicator does not distinguish between height and body mass in an age period where many children are experiencing the pubertal growth spurts and may appear as having excess weight (by weight-for-age) when in fact they are just tall" as indicated in Lines 154 to 160, and Lines 203 to 207. This means for WAZ, on 319 WAZ values were generated and they are for all children age 10 years and younger while for HAZ, 508 values for all children were generated.
Table 2 line 264, stunting changed to stunted in the third column.
Table 3 in line 266, all the values have been re-checked and percentages add to 100%. in addition, we tabulated frequency and percentages for sociodemographic variables of mothers. There is only one group in table 3, hence, there is no second group to compare with, no comparison test to use and no p-values generated.
Table 4 in line 268 and variables controlled for stunting are included in Lines 246 - 248, and for underweight in Lines 253 - 255.
Overweight/obesity mothers have been changed to mother who were overweight/obese, through out the documents in Lines 22-23, 307 and 312.
Clarity on the sentence "...40.2% and 75.3 % are lower prevalence rates than the 27 % found in the current study..." has be addressed by correcting lower to higher in Lines 320-321.
Reviewer 2 Report
Thank you for your work here. This topics extremely important globally and every country example thats shared is incredibly meaningful.
Prior to publication, I would recommend strengthening of your background information and scientific reasoning that was presented, in particular:
- the reasons why children are vulnerable: The reasons provided are not entirely accurate. Do not forget that they require less than adults, yet they are still growing and developing quickly (growth spurts, etc). They rely on others to provide food and drink and to teach them how to use it, etc.
- obesity is also undernutrition since most people who are obese also have micronutrient deficiencies.
- stunting began as an economic indicator, it is not simply chronic traditional undernutrition. (see recent works by Jef Leroy, Ed Frangillo, etc)
Author Response
All comments and additional information are written on blue in the manuscript:
we have strengthened the introduction and addressed vulnerability among children and growth spurts; Lines 73 -79.
We have read the article for Leroy and Frongillo, 2019, and added information on stunting.
The entire methods, results, discussion and conclusion were revised to fill up information where it was necessary.
Round 2
Reviewer 1 Report
The authors have responded to the previous comments and the manuscript has been improved. The following minor suggestions are recommended.
Line 18: Change “calculated” to “assessed”
Line 77: Apart from calories, general nutrients requirements also increase with the growth spurt. I suggest that the authors not restrict this to calories, rather use “calories and nutrients’ intake. Also, undernutrition is not only due to inadequate caloric intake but poor intake of other nutrients as well.
In table 1 authors must mention that WAZ was only calculated for children 10 years and younger in a footnote. This makes it clear that for the age group 10-15, subjects who were 11-15yrs old had no WAZ values.
Table 4: footnote is needed to describe what variables were controlled for in the models.
Author Response
The paper has been taken for English editing. Line 3 - we have added "System" in the title. It was an omission. Line 18 - "calculated" has been changed to "assessed" and it is highlighted in blue. Lines 80 and 82 - "calories and nutrients intake" has been added, highlighted in blue, and it is also covered by the references for those sentences. Line 266 - footnote has been added under table 1 and *asterisks put on WAZ variable within the table. Lines 272 -274 - footnote has been added and controlled variables for stunting indicated with one *asterisk and for underweight indicated with two **asterisks.